# Quality of Life of the Primary Caregivers of Children with Cleft Lip and Palate in Guanajuato, Mexico: A Cross-Sectional Study

**DOI:** 10.3390/healthcare12161659

**Published:** 2024-08-20

**Authors:** María del Carmen Villanueva-Vilchis, Karen Esperanza Almanza-Aranda, Luis Alberto Gaitán-Cepeda, Rubén Rangel-Salazar, María de los Ángeles Ramírez-Trujillo, Fátima del Carmen Aguilar-Díaz, Javier de la Fuente-Hernández

**Affiliations:** 1Department of Public Health, National School of Higher Studies, León Unit, National Autonomous University of Mexico, Leon 37684, Guanajuato, Mexico; cvillanueva@enes.unam.mx (M.d.C.V.-V.); kalmanzaa@enes.unam.mx (K.E.A.-A.); mramirez@enes.unam.mx (M.d.l.Á.R.-T.); faguilar@enes.unam.mx (F.d.C.A.-D.); 2Department of Oral Pathology and Oral Medicine, Graduate and Research Division, Dental School, National Autonomous University of Mexico, Mexico City 04360, Mexico; lgaitan@unam.mx; 3Department of Medical Sciences, Health Sciences Division, University of Guanajuato, León 37320, Guanajuato, Mexico; ruben.rangel@ugto.mx

**Keywords:** cleft lip, cleft palate, caregivers, quality of life

## Abstract

Comprehensive treatment is crucial for patients with a cleft lip/palate. While studies have investigated its impact on children’s quality of life, few have examined the effects on primary caregivers. The aim of the study was to compare the quality of life of caregivers of children with cleft lip/palate to a control group at the National School for Higher Studies, National Autonomous University of Mexico, Guanajuato. A cross-sectional study was conducted at a teaching dental clinic of the National School of Higher Studies, National Autonomous University of Mexico, Guanajuato, México, from May to December 2021 involving 140 caregivers (70 in each group). The WHOQoL Bref instrument assessed the quality of life. In addition to the descriptive analysis, a binary logistic regression analysis was carried out, taking dichotomized reported quality of life as the dependent variable. Among the caregivers, 88.6% were female (*p* > 0.05), and 45 (64.8%) from the study group reported poor quality of life (*p* < 0.05). A multivariate analysis indicated that caring for a child with cleft lip/palate raised the likelihood of poor quality of life (*p* < 0.05). The findings emphasize the need for comprehensive support for both patients and caregivers, as their well-being affects patient outcomes.

## 1. Introduction

Cleft lip/palate (CLP) is an abnormal congenital cleft that strongly affects the oral cavity and related structures [1]. Its multifactorial etiology includes genetic and epigenetic factors, especially environmental factors [1]. The worldwide incidence of CLP is 1:600/800 live births (1.42:1000), while isolated cleft palate (CPO) occurs in approximately 1 in 2000 births. Males are more affected than females, in a 2:1 ratio [2].

In Mexico, a prevalence of CPO/CLP of 5.3:10,000 births has been reported, mostly in males (6.3:10,000 births) compared to females (4.2 per 10,000 births) [3]. Likewise, in Guanajuato, Mexico, the observed prevalence was above the national rate of 6.2 per 10,000 births from 2008 to 2014 [3].

CLP deeply affects the psychological and social aspects of patients. In addition, CLP impacts at the family level by representing an eight-fold financial burden on healthcare costs. Also, primary caregivers must frequently attend medical appointments, resulting in absences from work and the need to change or postpone previously planned activities [4]. These factors can cause work-experience overload that reduces quality of life, leading to illness or stress-related conditions [5,6,7].

Nowadays, in the United States, the impact of caring for a person with CLP on social avoidance, distress, fear, self-esteem, and interpersonal support has been evaluated [4]. In Turkey [8,9] and India [10], the quality of life and stress of caregivers of children and adolescents with and without CLP have been assessed. In countries such as Brazil [5,11] and Nigeria [12], the relationship between quality of life and sleep quality and caregiver burden has been evaluated. However, no studies have been carried out in Mexico, although the state of Guanajuato has the highest prevalence of CLP in the country.

Since the perception of quality of life is self-perceived and dependent on particular social and demographic conditions, it is necessary to gather extensive data on various situations to better understand patients and their caregivers. Therefore, the purpose of the present study is to compare the quality of life of primary caregivers of children with CLP compared to caregivers of children of a control group at the dental clinics of the ENES León UNAM, Guanajuato during the year 2021.

## 2. Materials and Methods

### 2.1. Study Design

This is a paired cross-sectional study that was approved by the Research and Ethics Commission of the ENES León UNAM (CEI_21_04_S15).

### 2.2. Setting

ENES León offers a university program that provides comprehensive care for patients with orofacial clefts named “TiENES que sonreír, UNAMos esfuerzos” (you must smile, joint effort). The recruitment was performed from May to December 2021.

### 2.3. Participants

The primary caregivers of the patients in this program were invited to participate in the study. Primary caregivers were considered to be the ‘father’, ‘mother’, ‘uncle/aunt’, and ‘grandfather/grandmother’ who care most of the time for the child with a cleft. The study population consisted of 300 primary caregivers of patients with CLP who attended the above-mentioned program, while the control population consisted of 150 caregivers of children without CLP who participated in the pediatric dentistry clinic of the same institution.

The inclusion criteria considered caregivers to be included if they were living in the same house as the patient and provided primary care for them for at least 8 h per day. The exclusion criteria consisted of those primary caregivers with visual or hearing problems that prevented them from answering the questionnaire; these were excluded. The elimination criteria referred to those who did not answer the questionnaire in its entirety. The children’s data were obtained from clinical records, excluding those with incomplete or illegible information. The ages of the children were used as the matching criterion.

### 2.4. Variables and Data Source

After obtaining the signed consent form, a questionnaire on sociodemographic data, psychological variables, presence and characteristics of CLP, and quality of life was applied.

The sociodemographic variables included gender, level of education, occupation, marital status, and socioeconomic status using a ladder-shaped scale, with the lowest step representing low economic income and the highest step representing high economic income. The participants were positioned on the step that best represented them, and subsequently, a categorization was carried out, classifying them into three groups, namely low, medium, and high [13]. As a psychosocial variable, stress was measured through the Parenting Stress Index in its short Spanish version (α Chronbach = 0.91) [14].

The independent variable considered for the present study was the presence of a cleft lip/palate, a congenital abnormal gap that can occur in the upper lip, alveolus, or palate [1] and the dependent variable was the quality of life, a state of general well-being that includes objective descriptors and subjective assessments of physical, material, social and emotional well-being, along with the degree of personal development and activities, all weighted by a set of personal values [15]. The latter was assessed using the World Health Organization Quality of Life (WHOQoL-Bref) instrument, validated in Spanish [16]. WHOQOL-Bref shows a very high reliability value (Chronbach α = 0.895) and consists of 26 Likert-type questions, which were rated on a scale from 1 to 5, where 1 = never, 2 = almost never, 3 = sometimes, 4 = frequently, and 5 = almost always. The score obtained ranges from 26 to 130. The higher the score, the better the quality of life. The WHOQOL-Bref encompasses four domains, including aspects of physical health and ability to engage in different activities (7 items); psychological health related to body image and appearance (6 items); social relationships and social support resources (3 items); and environment involving the material and healthcare resources available (8 items). It also contains questions on the overall perception of quality of life and general health satisfaction. It should be mentioned that this variable was categorized into two according to the median of the quality-of-life scale.

### 2.5. Bias

As this was an age-paired study, the pairing may help to control for confounding bias on quality of life. So, the pairing was carried out according to the age of the children because, as has been reported, younger children may face greater feeding problems and may require a greater number of surgeries to correct the cleft [9].

### 2.6. Study Size

The sample size was calculated through the GRANMO Datarus application [17], considering a one-tailed hypothesis with the following assumptions: the probability of exposure in controls = 0.35, odds ratio = 2.5, and non-response bias of 0%. The value of OR = 2.5 was taken as a reference to that reported in the results by Médard et al. (OR = 3.0), considering the risk of having a poor quality of life in the family when there is a history of having an infant with CLP [18]. Thus sample size of 70 caregivers per group was determined and was chosen by convenience.

### 2.7. Statistical Methods

For statistical analysis, frequencies were obtained for the socio-demographic variables. Mean and dispersion were calculated for quantitative variables. Bivariate chi-square analyses were performed to verify differences in quality of life between the study groups. Finally, a binary regression–logistic model was used, entering variables that were significant in the bivariate analysis with a *p*-value ≤ 0.20. The significance threshold to enter the variables into the model is set according to the authors’ criteria and the theoretical relevance they consider the included variables to have. According to some authors, the significance value for multivariate models can range from 0.05 to 0.15 in the case of linear models and up to 0.20 in the case of binary models [19,20]. The quality of life was categorized into two according to the median of the quality-of-life scale.

Data from clinical records were used to evaluate clinical variables such as the type of cleft, affected site, and structures involved.

## 3. Results

One hundred forty caregivers participated in the study. Females were 88.6% (*n* = 124) (mean age 35.34 ± 9.4), and 11.4% (*n* = 16) were male (mean age 36.06; ±4.9). No statistically significant difference was observed according to the sex distribution (*p* > 0.05).

The results obtained when comparing the sociodemographic variables between the study groups are shown in Table 1. It can be observed that the caregivers in the control group had a higher level of education than the study group, since 54.3% (*n* = 38) of them had a high school level or higher (*p* = 0.003).

The mean age of all the children of both study groups was 8.8 ± 3.4 years, with 9.30 ± 3.3 years for girls and 8.36 ± 3.6 for boys. Table 2 shows that 60% (*n* = 42) of the patients with CLP were boys, while 51.4%. (*n* = 36) of the patients in the control group were girls. No statistical difference was observed in the sex distribution between the two groups (*p* = 0.175). Similarly, 80% (*n* = 112) of the mothers fulfilled the role of caregiver, regardless of the study group, with 84.3% (*n* = 59) in the CLP group and 75.7% (*n* = 53) in the control group. No statistically significant difference was observed according to the caregiver’s relationship by group (*p* = 0.39) (Table 2).

Regarding the clinical characteristics of patients with CLP, 44% (*n* = 31) presented with unilateral cleft lip and palate, 24% (*n* = 17) with isolated cleft palate, and 20% (*n* = 14) with bilateral cleft lip palate. The total number of cases is presented in Table 3. Eighty-seven percent (*n* = 61) of the patients reported having undergone one to three surgeries, 7% (*n* = 5) reported having undergone four or more surgeries, and only 6% (*n* = 4) had not undergone any surgery at the time of the present study.

Table 4 shows that mothers spent more hours per day caring for a child with CLP, with an average of 17.231 (SD ± 7.751) hours per day, compared to fathers who reported an average of 12.061 (SD ± 8.322) hours per day (*p* = 0.014).

Regarding the quality of life of female caregivers, 65.2% (*n* = 45) of caregivers of children in the control group reported a good quality of life, while 64.8% (*n* = 46) of caregivers of children with CLP reported a poor quality of life (*p* = 0.0001), as seen in Table 5.

No relationship was observed between reported quality of life, whether categorized as good or poor, and the number of surgeries undergone by their child (*p* = 0.893).

The analysis of the four domains of the WHOQoL Bref instrument showed that the physical health domain showed a significant difference between the two groups. The group of caregivers of control-group patients reported a higher score on the physical well-being domain than the caregivers of children with CLP (66.326 vs. 59.234, respectively; *p* 0.001). Similarly, caregivers of the control group reported higher scores in the environmental domain (*p* = 0.005). Table 6. 

A binary logistic model was carried out to evaluate the effect of the presence of cleft lip and/or palate on the quality of life of the caregivers (Table 7), in addition to other covariates such as stress and hours of care per day, which were significant with a *p* value ≤ 0.20. The results show that caregivers who have children with cleft lip and/or palate show a 3.2 times greater probability of having a poor quality of life, while those caregivers with clinical levels of stress show a 1.02 times greater probability of having a poor quality of life. This model explains 23.7% of the variation in quality of life, adjusted for the variables mentioned above.

## 4. Discussion

This study found that caregivers of patients with CLP are 3.2 times more likely to have a poor quality of life. As shown in Table 5, we found other variables that also modify caregivers’ quality of life, such as caregiver stress and hours of childcare. The latter is not well established as to why it is playing as a protective factor in the multivariate analysis. However, it may be a matter of reaching the highest point of the learning curve in caring for children with this type of abnormality.

The primary caregivers of children with dysmorphology, including CLP, play a crucial role as both psychological and financial supporters of the individuals under their care. Therefore, any imbalance in emotional and mental stability can affect the patient’s behavior [21]. The psychological status and the quality of life of the parents of patients with orofacial clefts, mainly the primary caregivers, are factors that impact. In the present study, a comprehensive assessment of quality of life was conducted using the WHOQoL Bref instrument, which allowed the analysis of physical, psychological, and social aspects of the primary caregivers of patients, both with and without CLP.

Our results reveal that mothers predominantly assumed the role of primary caregivers, regardless of the study group, which is consistent with previous reports in the scientific literature [12,22], according to which they dedicated 17 to 24 h a day to their care. Ribiero et al. [22] pointed out that a close relationship between mother and child is paramount for rehabilitating patients with CLP. In recent years, it has been reported [8] that fathers are also involved in childcare, although mothers typically remain more actively engaged in the primary caregiving role.

Given the above, it becomes evident that caring for children with CLP affects the primary caregivers’ lives and their immediate family environment in different ways. On the one hand, the hours invested in caring for infants can condition the entry of caregivers into the labor market, leading to school dropout and acceptance of part-time jobs and, consequently, leading to lower incomes [23]. These findings are consistent with our results, as they revealed that mothers, who were the primary caregivers of CLP patients, often had lower levels of education and were frequently unemployed. This contrasts with the control group, where unemployment rates were lower. This disparity may stem from the demanding nature of caring for a child with CLP, requiring round-the-clock availability to accompany the patient during treatment, particularly in the early years of life. The care of children with special healthcare needs can profoundly affect labor participation, which is directly proportional to the additional healthcare needs of the child [24]. Indeed, a study found that 80% of parents made employment decisions based directly on their child’s health status, with one-third eventually ceasing work to care full-time for their child with special healthcare needs [25]. On the other hand, medical care for CLP patients often requires long-term commitments. While children without special healthcare needs gradually become more independent, caregivers of cleft patients may face longer-term care responsibilities, limiting or even preventing their return to regular employment [24].

Regarding the quality of life reported by caregivers, our results show a significant difference between the two groups. Caregivers of patients with CLP exhibited poorer quality of life than the control group. These findings are consistent with the scientific literature, which indicates that quality of life is better in caregivers of patients without clefting [9,26]. Specifically, families with a member who has CLP report a significantly lower quality of life in the domains of physical, social, and psychological areas compared to control-group families, especially as the patient grows toward adolescence [9]. As proposed by Hatzmann et al., caregivers of chronically ill children have a low quality of life and are at risk of further decline [27].

In our study, caregivers who reported worse quality of life were those caring for patients who had undergone one to three surgeries and those who had patients between 7 and 11 years of age. Although children with CLP have completed lip and palate reconstructive surgeries before this age, aesthetic rehabilitations, including dental, orthodontic, or orthognathic treatments and speech development interventions, will continue during adolescence [9]. However, a recent study found no statistical difference in the total score and the four dimensions of quality of life among caregivers of patients with cleft lip and palate [21]. The quality of life of a patient’s parents is closely related to the patient’s physical and psychological characteristics [21].

Our results indicated that caregivers with lower educational levels had a worse quality of life than those who had high school or upper academic degrees, and those caregivers who were unemployed reported a poorer quality of life than caregivers who were employed professionals. Previous studies have identified cleft type as a significant factor associated with the quality of life of caregivers of patients with CLP [12,26]. Specifically, families with children who have isolated cleft palate tend to have a better quality of life than those with cleft lip/palate or cleft lip [26]. However, in our study, cleft type was not related to the quality of life of caregivers.

Mexico has a high prevalence of CLP cases, with Guanajuato being one of the states with the highest prevalence of CLP. Although there are studies that refer to the quality of life of children with CLP, this is the first study in our country that addresses the quality of life of caregivers despite its high prevalence. Therefore, our results emphasize the need to provide comprehensive treatments not only focused on children with this or other facial anomalies, but also on their caregivers and eventually promote greater well-being of the binomial.

### Limitations

The present study has some limitations, the first of which is that the results cannot be generalized, as the data were obtained from a population attending care clinics. On the other hand, the questionnaires used for the evaluation of both the stress scale and the quality of life scale are self-administered instruments, which could lead to a potential information bias.

## 5. Conclusions

Caregivers of patients with CLP are more likely to have a poorer quality of life than caregivers of children without the disease. The caregivers’ quality of life could be influenced by caregiver stress and hours of childcare. Other psychological characteristics of caregivers, such as stress, depression, and coping strategies employed, need to be further investigated. Finally, health-education approaches, such as motivational interviewing, should be further explored to identify perceived barriers to caregiving and their improvement, which will eventually improve the quality of life of this highly vulnerable dyad.

## Figures and Tables

**Table 1 healthcare-12-01659-t001:** Socio-demographic data of caregivers of patients attending the dental clinic of ENES León.

		CLP-Patient Caregivers		Control-Group Caregiver		Total		* *p*
		*n*	%	*n*	%	*n*	%	
Gender	Female	64	91.4	60	85.7	124	88.6	0.288
	Male	6	8.6	10	14.3	16	11.4	
	Total	70	100	70	100	140	100	
Level of education	Primary school	14	20	9	12.9	23	16.4	0.003
	Middle school	38	54.3	23	32.9	61	43.6	
	High school and upper	18	25.7	38	54.3	56	40	
	Total	70	100	70	100	140	100	
Occupation	Employee	21	30	27	48.6	48	34.3	0.016
	Unemployed	44	62.9	29	31.4	73	52.1	
	Professional	1	1.4	9	12.9	10	7.1	
	Other	4	5.7	5	7.1	9	6.4	
	Total	70	100	70	100	140	100	
Marital Status	Singles/Free Union	24	34.3	12	17.1	36	25.7	0.043
	Married	43	61.4	51	73.9	94	67.1	
	Divorced/Widowed	3	4.3	7	10	10	7.1	
	Total	70	100	70	100	140	100	
Socioeconomic status	Low	19	27.1	14	20	33	23.6	0.573
	Middle	49	70	53	75.7	102	72.9	
	High	2	2.9	3	4.3	5	3.6	
	Total	70	100	70	100	140	100	

CLP = cleft lip palate; * = chi-square.

**Table 2 healthcare-12-01659-t002:** Socio-demographic data of patients attending the dental clinic of ENES León.

		CLP Patient		Control-Group Patient		Total		* *p*
		*n*	%	*n*	%	*n*	%	
Gender	Female	28	40	36	51.4	64	45.7	0.175
	Male	42	60	34	48.6	76	54.3	
	Total	70	100	70	100	140	100	
Relationship	Father	6	8.6	11	15.7	17	12.1	0.390
	Mother	59	84.3	53	75.7	112	80	
	Other	5	7.1	6	8.6	11	7.9	
	Total	70	100	70	100	140	100	

CLP = cleft lip palate; * = chi-square.

**Table 3 healthcare-12-01659-t003:** Distribution by cleft type and number of surgeries received of CLP patients attending the program “TiENES que sonreír, UNAMos esfuerzos” ENES, León.

		Frequency	
		*n*	%
Cleft type	Unilateral cleft lip and palate	31	44
	Isolated cleft palate	17	24
	Bilateral cleft lip and palate	14	20
	Unilateral cleft lip	7	10
	Bilateral cleft lip	1	2
Number of surgeries	None	4	6
	One to three surgeries	61	87
	Four or more surgeries	5	7

**Table 4 healthcare-12-01659-t004:** Time spent by caregivers on patient care at the dental clinic of ENES León.

		*n*	Mean	SD	* *p*
How many hours a day do you spend caring for your relative?	Female	124	17.23	7.751	0.014
	Male	16	12.06	8.322	
	Total	140			

* = Student’s *t*-test.

**Table 5 healthcare-12-01659-t005:** Quality of life of caregivers of patients attending the dental clinic of ENES León.

		Good Quality of Life		Poor Quality of Life		Total		* *p*
		*n*	%	*n*	%	*n*	%	
Caregiver of patient with CLP and control group	Caregivers of control-group patients	45	65.2	25	35.2	70	50	0.0001
	Caregivers of patients with CLP	24	34.8	46	64.8	70	50	
	Total	69	100	71	100	140	100	

CLP = cleft lip palate; * = chi-square.

**Table 6 healthcare-12-01659-t006:** Quality of life across dimensions of the WHOQoL Brief instrument, as applied to caregivers of patients attending the dental clinic of ENES León.

		*n*	Mean	SD	* *p*
Physical Health	Control-group caregivers	70	66.326	15.496	0.001
	CLP-patient caregivers	70	59.234	8.884	
	Total	140			
Psychological Health	Control-group caregivers	70	64.345	16.524	
	CLP-patient caregivers	70	61.071	15.828	0.233
	Total	140			
Social relationships	Control-group caregivers	70	57.857	22.019	
	CLP-patient caregivers	70	55.00	14.286	0.364
	Total	140			
Environment	Control-group caregivers	70	57.053	16.838	
	CLP-patient caregivers	70	50.178	11.133	0.005
	Total	140			

CLP = cleft lip palate; * = Student’s *t*-test.

**Table 7 healthcare-12-01659-t007:** Final binary logistic regression model of the quality of life of the primary caregivers of patients attending the dental clinic of ENES, León.

Model Variables	*p*	OR	CI 95%	R^2^ Cox Y Snell
CLP/control	0.001	4.203	1.855–9.523	0.237
Stress	0.025	1.024	1.003–1.045	
Patient care hours (From 9 to 17 h per day)	0.021	0.232	0.067–0.804	
Patient care hours (From 18 to 24 h per day)	0.029	0.316	0.112–0.888	

## Data Availability

The data sets used and/or analyzed during the current study are available from the corresponding author upon reasonable request.

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
