# Peer review of "Quality of Life of the Primary Caregivers of Children with Cleft Lip and Palate in Guanajuato, Mexico: A Cross-Sectional Study"

_healthcare, 2024, doi:10.3390/healthcare12161659_

Round 1

Reviewer 1 Report

Comments and Suggestions for Authors

Title

-       Please improve the title by informing the study design and the study location.

Abstract

-       The research gap of the study was not presented in the abstract section. Please improve the novelty of the study in the abstract section.

-       The abbreviation should be outside the abstract section.

-       Please provide the location of the study.

-       Please provide the statistical analysis method.

-       Please provide a study setting.

-       The aim of the study was clear. However, it does not describe the study title. Please make the aim and the study title match.

Introduction

-       Please increase the scientific background of why the study topic must be conducted. Then, it describes what makes Mexico (I guess the study was conducted in Mexico since, as far as I read the title to the introduction, there was no information about the study location) different from other countries and presents the scientific reason why the study should be conducted in Mexico. Finally, this point will raise the study's novelty in the introduction. Please also provide the study novelty for the main variable in the introduction section. 

-       More, please raise the scientific reason why the study population was conducted among the caregiver of CLP. If the issue was low QoL, then why? what is the impact? Is that show an urgency? Why did the author compare the level of QoL with the control group (caregiver without CLP)) Please make its scientific reason be stronger.

-       Please have a better line throughout the manuscript from the title to the conclusion to match.

-       Once again, the title did not describe well the study result.

Method

-       Please provide the study size calculation reference, including the OR value.

-       Please provide clear inclusion and exclusion criteria for participants.

-       Please explain how to interpret the instrument.  Please also mention the validation or reliability value for the Spanish version.

-       Why did the author use the p-value <0.20? I recommend using 0.05 for the p-value.

-       I recommend using STROBE guidelines to report the study. Thus, the method section must be reported completely with these points:

1.     Study design

2.     Setting (location setting, relevant dates, including recruitment, exposure, follow-up, and data collection)

3.     Participants information

4.     The variable which is followed by the variable definition

5.     Data source

6.     The strategy to reduce bias

7.     Study size and how to calculate it

8.     How the quantitative variables were handled

9.     Statistical method

-       Please mention the value of the validity or reliability test of the instruments.

-       Please provide the complete details of how the author provides the data (n, %, mean, or SD?) in the statistical analysis.

Result

-       Please emphasize the study finding in early result section.

-       No table information, including the abbreviation information.  

-       In Table 1, there was a significant covariate. How did the author adjust it?

Discussion

-       Besides the main result, please raise the scientific problem about the variables among participants in Mexico and promote this study’s result as the strategy to solve its scientific problem.

-       The study utilized self-administered tools for data collection. This issue must be addressed to be the study’s limitations since this method might lead to a result bias.

-        

Reviewer 2 Report

Comments and Suggestions for Authors

Dear authors,

Many thanks for this interesting work. The topic is intriguing and not well researched and studies like the present one are welcomed to improve our knowledge and understanding of the medical and non-medical caregivers.

While the premise and foundation of the study are clear and strong, the presentation of methods and results need substantial clarifications and rewrite to improve the flow and content of the manuscript.

a. "The present study was a matched cross-sectional study". What is meant by the term "matched" in this sentence? I missed where the authors mention the use of propensity score matching to align their 2 groups. The authors do mention that "The age of the children was used as the matching criterion" however is their evidence to support that such precise matching is better than just taking a sporadic uniform age range for both groups?

b. "“TiENES que sonreir, UNAMos esfuerzo" - would it be possible for the authors to also provide a rough translation for the name of the program in brackets for English audience.

c. "The study population consisted of 300 primary caregivers of patients....." What is meant by primary caregivers? Please be specific and include examples of which professionals/relationships were considered as primary caregivers in the program.

d. "while the control population consisted of 150 caregivers of children who participated in the pediatric dentistry clinic of the same institution" - I assume the control population is the non-CLP caregivers? If so, again please be clear and consistent in terming the groups across the study.

e. "probability of exposure in controls = 0.35, OR= 2.5, and Non-response Bias of 0%" - why did the authors choose exactly these values? Authors should add references from previous studies that demonstrate these values or closely related values. Furthermore, its not clear which software was used to calculate sample size.

f. Assuming the conditions mentioned above, the authors have already assumed a one-sided hypothesis (since OR is not =1 but >1), hence all P values used must be 1 sided P values. The authors should clearly mention their hypothesis and the tail of significance used/justified under a separate header in the methods section.

g.  "Thus, a sample size of 70 mother-child pairs per group was determined" Better to phrase as sample size of 70 caregivers per group was determined.

h. " Finally, a binary regression-logistic model was used, entering variables that were significant in the bivariate analysis with a p-value ≤ 0.20." I missed where the results of the binary model are presented. Also why P < or = 0.20? If in your sample size you used confidence at 95% then why not use cutoff also at P < 0.05?

i. How was monthly economic income defined and categorized? Provide the salary ranges used and source of those ranges. 

Comments on the Quality of English Language

English can be improved and organization of the manuscript sections can be improved as well.

Reviewer 3 Report

Comments and Suggestions for Authors

Dear Authors,

Thank you for the opportunity to review your manuscript. The topic you have addressed is of significant importance, and your efforts in exploring the quality of life of primary caregivers of cleft lip and palate patients are commendable. Below are my detailed comments and suggestions aimed at improving the clarity, depth, and overall quality of your manuscript.

Abstract

  1. The aim of the study in the abstract needs to be expanded to clearly outline the specific objectives and the scope of the research.
  2. The conclusions should be expanded to provide a more comprehensive summary of the findings and their implications.
  3. Add a section on future research directions to highlight potential areas for further investigation.

Introduction

  • The introduction is well-written and provides adequate background information.

Materials and Methods

  1. The Materials and Methods section should be divided into multiple subsections to enhance readability and comprehension. Suggested subsections include:
    • Study Design and Participants
    • Data Collection
    • Ethical Considerations
    • Statistical Analysis
  2. Create a dedicated subsection for statistical analysis to provide clear and detailed information on the statistical methods used.

Results

  • When referring to percentages, please include the corresponding sample size (n) in the text for clarity and context.

Discussion

·      Please include a discussion of the limitations of your study. This will provide context for interpreting the results and understanding the potential constraints of your findings.

Example: "This study has several limitations. First, the cross-sectional design does not allow for causal inferences. Second, the sample size, although adequate for initial comparisons, may limit the generalizability of the findings. Finally, the reliance on self-reported measures may introduce response bias."

Conclusions

  1. The conclusions should be expanded to provide more details of the study findings and their implications for practice and policy.
  2. Include suggestions for future research to address gaps identified in the current study and to explore new areas of interest related to the quality of life of primary caregivers of cleft lip and palate patients.

Your manuscript addresses a critical area of research, too often overlooked, and with the incorporation of these suggestions, it can significantly enhance its impact and readability. I look forward to seeing the revised version and commend you on your important contribution to this field.

Best regards!

Round 2

Reviewer 2 Report

Comments and Suggestions for Authors

Dear authors,

Thank you for addressing my concerns. The revised manuscript is much more clearer and easy to follow. I look forward to reading the published version online.

Comments on the Quality of English Language

Minor changes maybe undertaken.